# Exploring the Effect of the Irradiation Time on Photosensitized Dendrimer-Based Nanoaggregates for Potential Applications in Light-Driven Water Photoreduction

**DOI:** 10.3390/nano9091316

**Published:** 2019-09-14

**Authors:** Natalia P. Martínez, Ricardo Inostroza-Rivera, Boris Durán, Leonard Molero, Sebastián Bonardd, Oscar Ramírez, Mauricio Isaacs, David Díaz Díaz, Angel Leiva, César Saldías

**Affiliations:** 1Departamento de Inorgánica, Facultad de Química y de Farmacia, Pontificia Universidad Católica de Chile, Macul, 7820436 Santiago, Chile; 2Facultad de Ciencias de la Salud, Universidad Arturo Prat, 1100000 Iquique, Chile; 3Centro de Investigación en Nanotecnología y Materiales Avanzados, Pontificia Universidad Católica de Chile, 7820436 Macul, Chile; 4Departamento de Química Física, Facultad de Química y de Farmacia, Pontificia Universidad Católica de Chile, Macul, 7820436 Santiago, Chile; 5Centro de Nanotecnología Aplicada, Facultad de Ciencias, Universidad Mayor, 8320000 Santiago, Chile; 6Institut für Organische Chemie, Universität Regensburg, Universitätsstr. 31, 93053 Regensburg, Germany or; 7Instituto de Productos Naturales y Agrobiología del CSIC, Avda. Astrofísico Francisco Sánchez 3, 38206 La Laguna, 38206 Tenerife, Spain

**Keywords:** dendrimers, photoactive, hydrogen evolution, artificial photosynthesis

## Abstract

Fourth generation polyamidoamine dendrimer (PAMAM, G4) modified with fluorescein units (F) at the periphery and Pt nanoparticles stabilized by L-ascorbate were prepared. These dendrimers modified with hydrophobic fluorescein were used to achieve self-assembling structures, giving rise to the formation of nanoaggregates in water. The photoactive fluorescein units were mainly used as photosensitizer units in the process of the catalytic photoreduction of water propitiated by light. Complementarily, Pt-ascorbate nanoparticles acted as the active sites to generate H_2_. Importantly, the study of the functional, optical, surface potential and morphological properties of the photosensitized dendrimer aggregates at different irradiation times allowed for insights to be gained into the behavior of these systems. Thus, the resultant photosensitized PAMAM-fluorescein (G4-F) nanoaggregates (NG) were conveniently applied to light-driven water photoreduction along with sodium L-ascorbate and methyl viologen as the sacrificial reagent and electron relay agent, respectively. Notably, these aggregates exhibited appropriate stability and catalytic activity over time for hydrogen production. Additionally, in order to propose a potential use of these types of systems, the in situ generated H_2_ was able to reduce a certain amount of methylene blue (MB). Finally, theoretical electronic analyses provided insights into the possible excited states of the fluorescein molecules that could intervene in the global mechanism of H_2_ generation.

## 1. Introduction

Modern civilization is highly dependent on fossil fuels, a nonrenewable energy source originally provided by the storage of solar energy [1,2,3]. Fossil-fuel dependence has severe consequences, including energy security issues and greenhouse gas emissions [4,5]. This constitutes an important threat to the human race as we know it due to the imminent depletion of traditional energy sources and contamination problems [6,7]. Thus, the international community has focused its attention on the discovery of alternative energy sources that can meet growing worldwide energy demand [8,9]. Many different alternatives have been envisioned to provide a sustainable solution to these issues. Solar energy has arisen as a promising alternative because it is an abundant, clean, highly available, and renewable type of energy resource available for use [10,11,12]. This potential is demonstrated by the effect of the radiant energy emitted by the sun on natural photosynthesis processes, demonstrating the feasibility of efficiently using sunlight to produce the photoinduced cleavage of the chemical bonds of water molecules [13,14,15]. Considering this, gaining a better understanding of natural photosynthesis at the molecular level has attracted much attention lately in order to create artificial photosynthetic systems having photoactive and donor–acceptor compounds [16,17,18,19]. Indeed, a significant amount of research effort has been directed toward the development of artificial systems composed of covalently attached molecules [20,21] or supramolecular arrangements [22,23]. According to this approach, macromolecular structures with well-defined architectures and monodispersity emerge as relatively viable and interesting alternatives for the development of viable photoactive systems [24,25,26]. Specifically, the use of dendrimers seems to be an ideal strategy due to their unique chemical structure and properties, such as compact globular shapes and sizes, significant peripheral functionalities, and appropriate stability in diverse chemical environments [27,28,29]. Importantly, the high functionality of polyamidoamine (PAMAM) provides a profitable way to covalently attach photosensitizers onto their numerous peripheral functional groups. For example, PAMAM dendrimers have been demonstrated to be a potentially useful photocatalytic platform for the water-splitting reaction using by photosensitizers such as ruthenium and iridium complexes [30,31]. Thus, photoactive dendrimers combined with metal complexes or nanoparticles acting as co-catalysts, along with an appropriate reducing environment, can give rise to the H_2_ evolution stage [32]. In this context, the water-splitting reaction performed on the outer surface of dendrimeric aggregates (i.e., in order to utilize the high number of functional groups) in aqueous environments could be a crucial aspect to increase the feasibility and efficiency of this process [30,33]. Thus, the employment of dendrimer-based nanoaggregates as a photosensitized system could lead to the facilitation of photocatalytic processes for applications in artificial photosynthesis. It would be possible to mimic natural photosynthesis, thus directly converting solar energy to fuel as hydrogen. In this work, we report the preparation of dendrimer-based nanoaggregates using a fourth generation polyamidoamine dendrimer that was chemically modified with fluorescein units at the periphery and Pt nanoparticles as the co-catalytic center intended to propitiate the water photoreduction reaction. It should be noted that the formation and characterization of diverse types of macromolecule–chromophore (e.g., fluorescein) conjugates is well-documented in the literature [34,35,36]. Thereby, these photosensitized systems were successfully employed for the in situ generation of H_2_ from an aqueous solution using a light-driven process. To the best of our knowledge, this is one of the few reports in the literature on water photoreduction that includes an adequate characterization of the behavior and properties of photosensitized PAMAM nanoaggregates with different irradiation times. The approach of this study gains more insights into the crucial aspects, namely, the stability and photoactivity, of these systems over time.

## 2. Experimental Section

### 2.1. Materials

Fourth generation amine-terminated polyamidoamine (PAMAM-NH_2_ G4 14,200 g/mol), fluorescein 5-isothiocyanate (>90%), sodium L-ascorbate (>98%), K_2_PtCl_4_, and methyl viologen dichloride hydrate (>98%) were purchased from Sigma-Aldrich (Saint Louis, MO, USA) and were used without further purification. Anhydrous methanol solvent was purchased from Merck (Kenilworth, NJ, USA). The water used in both the irradiation and photoreduction experiments was purified using a Millipore Milli-Q system (resistivity higher than 18.2 MΩ∙cm, Santiago, Chile). 

### 2.2. Synthesis of the PAMAM-Fluorescein Dendrimer

To 10 mL of 5% (*w*/*v*) PAMAM methanol solution, a solution of 1.0 mg/mL fluorescein 5-isothiocyanate in dry methanol was slowly added under vigorous stirring. The final concentration of fluorescein in the reaction medium was adjusted to eventually obtain an amine group:fluorescein ratio of 3:1. The reaction was kept in the dark at room temperature for 2 h. The obtained product was precipitated in tetrahydrofuran and then dried and washed several times with a Milli-Q water:methanol mixture (1:1) to remove the free fluorescein [37,38]. Finally, the PAMAM-fluorescein dendrimer was dried under vacuum and stored at 5 °C until further use. Using a titration method for the determination of free-amine groups after the reaction [39], the amine group:fluorescein ratio was determined to be approximately 5:1. It is worth mentioning that this ratio was the highest that could be obtained for this dendrimer under the described experimental conditions.

### 2.3. Preparation of Dendrimer Nanoaggregates 

To prepare the PAMAM-fluorescein nanoaggregates (G4-F NG), the G4-F dendrimer units were placed into an aqueous solution (1.5 × 10^−4^ mg/mL concentration), and subsequently, this solution was ultrasonicated for 3 h in order to promote the self-assembly process. This stage was characterized by the appearance of slight turbidity, which could be observed by the naked eye. Finally, the suspension was placed into a cellulosic dialysis system (in order to remove the free G4-F dendrimer units, Sigma-Aldrich (Saint Louis, MO, USA) with a molecular mass cut-off of 20 kDa under stirring for 24 h against a methanol:water (80:20) mixture. The medium was replaced every 4 h. After dialysis, G4-F NG were concentrated using centrifugation, collected, and dried under vacuum for 24 h.

### 2.4. Synthesis of Pt-Ascorbate Nanoparticles

The synthesis of Pt nanoparticles stabilized using sodium L-ascorbate was performed based on a previously reported method with some modifications [40]. A previously mixed aqueous solution of 0.05 M K_2_PtCl_4_ and 0.1 M sodium L-ascorbate was added to an aqueous solution under reflux with stirring at 353 K. The color of the solution was observed to turn from pale yellow to black. After 3 h, the reaction was stopped by immersing the flask in cool water.

### 2.5. Equipment for Characterization 

Absorption measurements were recorded with an Agilent Cary 60 spectrophotometer between 400 nm and 800 nm at 25 °C. Fourier Transform infrared spectroscopy (FT-IR) spectra were recorded on Nicolet iS10 instrument (Waltham, MA, USA). Dynamic light scattering (DLS) measurements were performed using a Nicomp 380 particle size analyzer (Billerica, MA, USA) equipped with an Ar-ion laser. Dust-free vials were used, and measurements were performed at 298 K at an angle of detection of 90°. Polarized light microscopy (PLM) images of the samples were obtained using an Olympus BX60 microscope (Tokyo, Japan) and a QImaging MP5 CCD digital camera. The samples were placed between two parallel microscope slides. The optical images were recorded and analyzed via the QImaging QCapturePro software (version 6.0). The morphology of previously selected samples was characterized after uranyl acetate staining using a Philips Tecnai 12 (Hillsboro, OR, USA) transmission electron microscopy (TEM) operating at 80 kV. The scanning Kelvin probe force microscopy (SKPFM) measurements were performed at room temperature by a M470 workstation using a SKP470 scanning Kelvin probe (Willow Hill, PA, USA) with a vibrating capacitance probe (proprietary SKPR tungsten air gap).

### 2.6. Irradiation and Water Photoreduction Experiments

Both irradiation and water photoreduction experiments were performed in carefully sealed headspace vials with controlled stirring. The 5 mL sample solutions were degassed using a constant flow of argon gas for 30 min at room temperature prior to irradiation. Unlike in the irradiation experiments, to carry out the light-driven water photoreduction reaction, sodium L-ascorbate, methyl viologen, and Pt-ascorbate nanoparticles were added to be used as the sacrificial electron donor, electron relay agent, and co-catalyst, respectively. A 500 W Xe lamp (Santiago, Chile) with a filter cutting off the UV (*λ* ≤ 420 nm) and infrared light was used as the visible light source. The H_2_ generated from the water photoreduction reaction was characterized by GC analysis using a DANI MASTERS GC (Cologno Monzese, Italy) with a fused silica capillary column (SupelcoMol Sieve 5A plot, 30 m × 0.53 mm) coupled with micro thermal conductivity detector (μTCD) using argon as the carrier gas and methane as an internal standard.

### 2.7. Photogenerated H_2_ for the Reduction of Methylene Blue (MB) 

The reduction of MB was performed as a model reaction to assess the feasibility of the use of in situ photogenerated H_2_ as a reducing agent. Typically, 50 μL of a 1.0 mM MB solution was added into an aqueous solution of G4-F NG, Pt-ascorbate nanoparticles, sodium L-ascorbate, and methyl viologen. The change in the absorbance of the dye was monitored using UV-vis spectroscopy in the range of 660–670 nm (the wavelength of the maximum absorption of MB).

### 2.8. Computational Details

All of the calculations were performed at the density functional theory (DFT) level in the Gaussian 09 [41] software (version Gaussian 09; Carnegie Mellon University, USA). The hybrid B3LYP [42,43] exchange-correlation functional and the 6-311+G(d,p) [44] basis set were employed, and the quality of the minimization procedure was checked by verifying the absence of vibrational normal modes with imaginary frequencies. Geometrical optimization of all systems was performed at both the ground (S_0_) and the first excited state (S_1_). The latter was carried out using time-dependent (TD)-DFT [45] optimization as implemented in Gaussian 09. Implicit continuum solvent effects were included by the polarizable continuum model (PCM) [46] with water as the solvent (*ε* = 78.3553 D).

## 3. Results and Discussion

PAMAM functionalized with fluorescein (G4-F dendrimer units) was synthesized to obtain a conjugated dendrimer–chromophore in order to explore the stability and potential photosensitizer behavior over time for light-driven water photoreduction aimed at producing H_2_. The UV–vis spectra of bare PAMAM, free fluorescein, and PAMAM functionalized with fluorescein, as well as the proposed mechanism of the chemical reaction to form the bonds between fluorescein molecules and the primary amine groups located at the periphery of PAMAM, are shown in Figure 1.

For free fluorescein and G4-F dendrimer units, the absorbance maxima were located at approximately 460 nm and 480 nm, respectively. The differences in the shape of the absorption bands and wavenumber shifts were attributed to the covalently bonded PAMAM-fluorescein system and the influence of the environment that surrounds the fluorescein molecules (e.g., the presence of water molecules and interactions among fluorescein entities). 

### 3.1. Estimation of the Critical Aggregation Concentration (CAC) of G4-F NG 

To estimate the critical aggregation concentration using the DLS technique, aqueous solutions of G4-F dendrimer units were prepared within the concentration range of 1 × 10^−4^ to 1 × 10^−8^ mg/mL. The intensity values of scattered light as a function of the concentration of G4-F dendrimers are shown in Figure 2. The scattering intensities detected for G4-F dendrimer concentrations below the CAC had an approximately constant value corresponding to that of Milli-Q water. The intensity started to show a linear increase with concentration at the CAC because the number of dendrimer nanoaggregates (TEM micrographs are shown below) is increased in the aqueous medium. The intersection of the best fit lines drawn through the data points corresponds to 3.5 × 10^−5^ mg/mL, which represents the CAC value of the G4-F dendrimer. The obtained results were also confirmed by the autocorrelation function curves, which were similar to that of Milli-Q water at concentrations below the CAC. As the CAC was reached, the recorded autocorrelation function curves increased, thus reflecting that the intensity of scattered light increased due to the presence of vesicular aggregates. The size distribution of G4-F nanoaggregates above the CAC (1 × 10^−4^ mg/mL) at 25 °C was also estimated using DLS. The hydrodynamic diameter of dendrimer nanoaggregates, D*_h_*, was determined to be 53.3 ± 5.9 nm in aqueous medium. It should be noted that the D*_h_* did not change significantly at concentrations above the CAC.

### 3.2. UV-Visible Study

To investigate the behavior of the optical, structural, electronic, and morphological properties, as well as gain insights into the stability of prepared dendrimer nanoaggregates in an aqueous medium, irradiation experiments for different times were carried out. Importantly, it is highly relevant to unveil the stability of dendrimer nanoaggregates with different irradiation times, which also leads to achieving a better understanding of their potential photoactivity over time in the reaction medium. Initially, the absorption behavior would be indicative of the photosensitizer properties that could eventually contribute to the light-harvesting ability of the dendrimer nanoaggregates. Thus, possible cooperative effects between the covalently-bonded fluorescein molecules were expected to appear due to the self-assembling ability of dendrimers. Additionally, it was thought that these nanoaggregates would provide an adequate environment, robust photosensitized scaffold, and chemical stability to the chromophores under light irradiation in an aqueous medium. Keeping this in mind, a series of UV-visible spectra were recorded after different irradiation times to preliminary assess the behavior of these systems (Figure 3). The obtained spectra showed that during ≈12 h of continuous irradiation, the optical properties of the dendrimer nanoaggregates were not dramatically modified; however, a significant change in the absorbance after 12 h was clearly noted. A notable decrease of the absorbance was produced, which was likely due to drastic changes in the surrounding chemical environment of fluorescein. These changes could be related to disaggregation phenomena of the G4-F NG and eventually the subsequent cleavage of the PAMAM-fluorescein bonds, which contributed to the course of this light-induced irreversible process. Moreover, the apparent shift of the absorbance maximum and decrease of the absorption of the band located at 450 nm at 14 h of irradiation was observed. This was evidence that the fluorescein entities were possibly undergoing a photodecomposition reaction. It was likely that a photosensitized, relatively ordered, and rigidized environment was more favorable for the triggering of the photodegradation processes of the dendrimer nanoaggregates after an extensive time of continued irradiation. The images of the dendrimer nanoaggregates in aqueous medium after different irradiation times are shown in Figure 3b.

With the aim of estimating the energy threshold for photons to be absorbed, the optical band gap energy of dendrimer nanoaggregates was calculated as follows:(1)$$E_{g}=\frac{1242}{\lambda_{th}}$$
where $E_{g}$ represents the optical energy band gap (in eV) and $\lambda_{th}$ represents the threshold wavelength (in nm) obtained from the onset of the absorption spectrum. The above mathematical expression is usually valid for organic π-conjugated systems, i.e., fluorescein entities. According to the UV-visible spectra, the estimated $E_{g}$ for dendrimer nanoaggregates was approximately 2.3 eV. 

It should be noted that this entire part of the study motivated exploration of the behavior of dendrimers using the other techniques, as detailed below.

### 3.3. FT-IR Analysis

The effect of the irradiation time was also investigated using the FT-IR normalized spectra of G4-F NG samples (Figure 4). The representative FT-IR spectrum for an elapsed time of 12 h exhibited peaks belonging to the OH, C=O, and C=C stretching vibrations of the hydroxyl, carbonyl, and phenyl groups, which were observed at approximately 3470 cm^−1^, 1740 cm^−1^, and 1490–1380 cm^−1^, respectively. These bands were mainly attributed to the presence of fluorescein. This confirmed that PAMAM was successfully conjugated with fluorescein, hence forming part of the nanoaggregates. These results, along with those of the UV-visible spectra shown above, support our hypothesis that PAMAM was successfully conjugated to fluorescein entities. Notably, after 13 h of irradiation, the spectrum of dendrimer nanoaggregates showed changes in the shape of the bands and relatively moderate wavenumber shifts compared to that after 12 h of irradiation. This could have resulted from changes in the intermolecular interactions among the molecular entities that were constituting the aggregates. Therefore, these interactions could reflect variations in the vibrational modes of the chemical bonds involved in this process. It is likely that the photoactivation of chromophore functional groups of G4-F NG significantly contributed to this type of process. Complementarily, the spectrum at 16 h of irradiation showed a dramatic change in the shape and transmittance of characteristic bands, helping to corroborate the potential physicochemical disintegration (e.g., disaggregation of the dendrimers and cleavage of PAMAM-fluorescein bonds) and further photodegradation of the entities that composed the G4-F dendrimer nanoaggregates.

### 3.4. Surface Potential Analysis

SKPFM was employed as a useful technique for obtaining information on the potential electronic surface activity of the G4-F NG as a function of the irradiation time. The obtained surface potential images of the G4-F NG after different irradiation times are shown in Figure 5. The green and blue colors indicate relatively electronically passive regions, whereas the yellow and red colors represent electronically active zones. The passive regions were ascribed to the possible absence of photoactive zones, which were interpreted as having worse electronic photoexcitation phenomena than the active regions. These active regions would be composed of fluorescein domains that revealed significant electronic surface activity after irradiation. Thus, the presence of fluorescein-rich zones, as well as the relatively adequate distribution on the surface of the sample, were revealed. When G4-F NG were irradiated for 12 h, the surface potential was clearly higher in comparison to those samples irradiated for longer. The electronic activity for the sample irradiated for 12 h displayed a better distribution compared to other samples. After that time, an important decrease of the electronic activity could be noted. It is well-known that fluorescein exhibits important electronic activity that is mainly stimulated by light due to its chromophore behavior, which in turn, is also dependent on the physicochemical environment of this molecule. Additionally, the relatively distinguishable passive and active regions could be indicative of the ability of dendrimer–chromophore systems to facilitate and stabilize eventual charge transfers or separation phenomena. 

### 3.5. Polarized Light Microscopy Analysis

Polarized light studies for the G4-F NG were also conducted. Optical reflectance images obtained using PLM for the analyzed samples are presented in Figure 6. As can be observed, the presence of relatively small bright regions for G4-F NG were detected after 12 h of irradiation. Conversely, the PLM images for longer irradiation times lacked recognizable bright regions. Moreover, the complete disappearance of both light and dark regions was observed for the samples irradiated for 16 h. The above would indicate a typical amorphous behavior, which is in agreement with the chemical structure of dendrimers. Globally, dendrimer macromolecules did not display remarkable crystalline behavior; therefore, the visualization of these regions could be attributed to those wherein fluorescein molecules were present. This suggests that this chromophore eventually played a key role in the crystallinity/amorphous behavior of the G4-F NG depending on the period of time that it was exposed to light radiation. Considering this, that effect would be attributable to the inherent photoresponsive behavior of fluorescein over time. During the first stage, this behavior would cause a rearrangement at the molecular level, and subsequently, a change in the crystalline behavior at the local level as the time of exposure to light irradiation was longer, i.e., 16 h. It is likely that the presence of photoexcited fluorescein molecules promoted some degree of reorganization at the supramolecular levels (12 h of irradiation) of G4-F NG. Therefore, for longer light exposure times, the samples displayed a distinguishable decrease of the crystalline domains.

### 3.6. Morphological Analysis

Remarkably, some authors have reported the preparation of different morphologies of dendrimer aggregates based on PAMAM peripherally that was modified with different types of photoactive molecules [34,35,39]. Thus, these dendrimer–chromophore systems in aqueous medium can self-assemble into nanoaggregates due to the incorporation of hydrophobic chromophores, even at the low generations of dendrimers. Therefore, the peripheral chemical modification of PAMAM G4 with fluorescein molecules also leads to the formation of nanoaggregates by the self-assembly of PAMAM-fluorescein units in an aqueous solution. Figure 7 shows the obtained TEM micrographs and the histograms of the size distribution for the G4-F NG exposed to light irradiation for different times. The micrographs show appreciable variations of the morphology depending on the irradiation time of the sample. These variations were more marked in the case of the size distribution of the analyzed samples. With respect to this, as the irradiation time increased, the size distribution and the respective polydispersity tended to increase. The standard deviations for the samples irradiated for 12 h, 13 h, and 16 h were 26.5%, 45.4%, and 37.0%, respectively. According to this, the light response phenomena of fluorescein molecules should propitiate a marked change in the molecular mobility in the vicinity of the dendrimer periphery. Complementarily, the differences in the size distribution and standard deviation of the analyzed samples could be explained based on the eventual increase of the segment mobility of dendrimer nanoaggregates. This should affect both the regular morphology and mean diameter of the dendrimer nanoaggregates. According to the structure, functionality, and molecular organization of aggregates, fluorescein molecules would exhibit a certain rigidity (restricted mobility) when they were exposed to an aqueous environment. After irradiation, these hydrophobic zones were expected to be significantly more exposed to an aqueous medium, which could be ascribed to the possible higher viscosity of the microenvironment. The above would support the variations of the nanoaggregate morphologies and size distribution attributed to the light response of fluorescein with different irradiation times.

### 3.7. Water Photoreduction Performance

For these experiments, the platinum nanoparticles were prepared according to the above-mentioned method. The TEM micrograph with the respective size distribution of the obtained nanoparticles is shown in Figure 8. The reduction of potassium tetrachloroplatinate (II) in aqueous solution using different types of reducing agents (similarly to the route proposed by Turkevich et al. using citrate and KAuCl_4_ salt [47]) is one of the most widely used methods for generating Pt nanoparticles [48,49,50]. The overall reaction for the reduction of PtCl_4_^2−^ using L-ascorbate could take place as follows:PtCl_4_^2−^ + L-ascorbate + H_2_O → Pt^0^ + L-dehydroascorbate + H^+^ + Cl^−^

Thus, L-ascorbate would act as suitable reducing and protecting agent for the synthesis of Pt nanoparticles in an aqueous medium. 

The photocatalytic behavior of G4-F NG was examined in accordance with highly relevant reported works by Xun et al. [30] and Ravotto et al. [51]. The photocatalytic activity of the dendrimer nanoaggregates was analyzed using G4-F as a photosensitizer, sodium L-ascorbate as a sacrificial electron donor, Pt-ascorbate as a co-catalyst, and methyl viologen as an electron relay agent. The formation of hydrogen gas was detected after continuous irradiation with visible light from the previously purged aqueous suspension of G4-F NG (≈100 mg) with argon and Pt-ascorbate nanoparticles (≈20 mg) in the presence of 2.5 × 10^−3^ M sodium L-ascorbate and 5.0 × 10^−4^ M methyl viologen. According to the literature, the experiments were conducted in the pH interval of 5.5–6.5 to avoid both the protonation of free −NH_2_ groups [52] and undesirable effects on the photofunctional behavior of fluorescein [53]. As control procedures, free fluorescein (≈10 mg) was used as a photosensitizer under the same experimental conditions; however, a noticeable amount of hydrogen production was not detected. This evidence seems to suggest that an adequate solubility, stability, and photoactivity in the medium, as well as eventual intimate contact between the photosensitizer and the co-catalyst, is required for efficient hydrogen generation. It seems the photosensitized periphery of dendrimer nanoaggregates could act as a convenient zone of contact for the Pt-ascorbate nanoparticles, likely promoting or at least facilitating the process of electron transfer. Therefore, a necessary condition is for the photosensitized sites in the dendrimer nanoaggregates to be close enough to the catalytic center (i.e., Pt-ascorbate nanoparticles) in order to contribute to the appropriate hydrogen generation in the reaction medium. Importantly, the turnover number (TON) for the photosensitized dendrimer aggregates is defined as the moles of H_2_ evolved per mole of G4-F dendrimer units. Figure 9 summarizes the hydrogen evolution monitored over time. The photocatalytic activity ceased after approximately 10 h of irradiation, achieving up to 380 μmol of hydrogen. According to the amount of G4-F dendrimer used, this corresponded to an average value of 7.0 × 10^−6^ mol of PAMAM-fluorescein dendrimer, and thus, the turnover number (TON) was estimated to be 54 per G4-F dendrimer. The maximum rate of hydrogen production, calculated in the linear part of the kinetic curve, was 60.9 μmol h^−1^, which corresponded to a turnover frequency (TOF) of 8.7 h^−1^ per G4-F dendrimer. According to these results, a possible mechanism responsible for the formation of hydrogen is proposed in Scheme 1. The photosensitized G4-F NG under irradiation would be driven to an excited state (G4-F*). In the presence of ascorbate as an electron donor agent, it could be reductively quenched yielding the chemically reduced specie (G4-F*^−^). Thus, the adequate electron-accepting and electron-transporting properties of methyl viologen (V) allowed the electrons to be transferred from G4-F*^−^ nanoaggregate species to the catalytic surface of Pt-ascorbate nanoparticles. Therefore, in that site, H_2_ evolution occurs, while the initial excited state (G4-F*) returned to the ground state, propitiating the photoreduction of water [54,55].

### 3.8. Reduction of Methylene Blue Using In Situ Generated H_2_

A potential application of the in situ generated H_2_ was studied using a well-documented reaction model, specifically, the reduction of methylene blue (MB) [56,57]. The evolution of this reaction was monitored using UV-vis spectroscopy (Figure 10). This type of catalytic reaction could have potential in the discoloration of dye wastewaters, for example [58]. Considering the amount of wastewater containing these compounds that are discharged into water sources by diverse industrial activities, and subsequently, the difficulty in removing the compounds from the water, the catalytic reduction or discoloration of dye molecules has emerged as a very attractive treatment approach [59,60]. The initial UV-vis spectrum of the MB in the aqueous solution in the presence of the G4-F nanoaggregates, Pt-ascorbate nanoparticles, ascorbate, and methyl viologen showed a maximum absorption at approximately 665 nm. Interestingly, the intensity of this absorption band decreased after different irradiation times, which would indicate that the photocatalytic-generated H_2_ produced the reduction of MB. However, a marked decrease of the absorption of MB was no longer detected after five hours of irradiation. This behavior could be explained by the apparent inhibition of the generation of H_2_ in the medium. It is likely that the presence of a certain amount of the reduced species of MB (MBH_2_) favored either the quenching processes of the photosensitized dendrimer nanoaggregates or the passivation of the Pt nanoparticle surface to continuously produce H_2_ [61,62]. After irradiation treatment of MB, its remaining concentration was determined to be approximately 4 μM (the initial concentration in the reaction medium was 0.01 mM). This concentration was estimated using a calibration curve built with standard solutions of MB. It should be noted that diverse control experiments were performed by irradiating MB in the presence of ascorbate and methyl viologen (without dendrimer nanoaggregates and Pt-ascorbate nanoparticles), as well as in the presence of dendrimer aggregates and Pt-ascorbate nanoparticles (without ascorbate and methyl viologen) for the reduction reaction of MB. For all of the mentioned cases, no significant reaction was identified and no change at 665 nm was detected for MB in the UV-vis absorption spectrum. Therefore, the correct set up of the experiment, along with the ability of the global system to generate in situ H_2_ in the aqueous medium, played a key role in the process of MB reduction. 

### 3.9. Theoretical Analysis

To provide a better understanding of the experimental results obtained for the G4-F nanoaggregates, TD-DFT calculations were performed to generate the first 10 excited states due to singlet–singlet electron transitions from the S_0_ ground state in water as a solvent. It should be noted that an important limitation in the calculation of the electronic structures with a considerable number of atoms is the computational cost. Keeping in mind this limitation, the fluorescein entity (Figure 11) was chosen as a molecular model to mimic the real systems with the intention of reducing such costs. Table 1 shows the maximum energy (eV), monoexcitation (MO), contribution to oscillator strength, and oscillator strength (O Str). The optical transition at approximately 3.91 eV involved one transition, which according to Table 1, was the following: H-1 → L with 97% of the 0.0341 oscillator strength and a transition energy 3.91 eV. The single state at 4.21 eV had contributions from two transitions, the most important one being H-1 → L+1, with 76% of the 0.1421 oscillator strength and a transition energy of 4.21 eV. The single state at 4.27 eV had contributions from four transitions, but only one has terms that contributed significantly to the oscillator strength, which was H-2 → L, with 71% of the 0.1503 oscillator strength and a transition energy of 4.27 eV. Finally, the single state at 4.49 eV had contributions from four transitions, with the most significant transition being H-2 → L+1, with 70% of the 0.4717 oscillator strength and a transition energy of 4.49 eV. Figure 12 shows the TD-DFT calculated UV-vis optical absorption spectrum of fluorescein in water (PCM model). 

Figure 13 displays the four most important TD-DFT calculated contributions for the excited states and the molecular orbitals involved in the main optical absorption bands involved in the optically active transitions. Globally, it was observed that the nature of the excitation energies was in agreement with π →π∗ transitions, wherein the π orbital (i.e., HOMO -1 and HOMO -2) was delocalized over the entire structure and the π∗ orbital (i.e., LUMO and LUMO +1) was mainly localized on the 2-benzoforanone ring and isothiocyanate group. This allowed it to be proposed that the fluorescein entities in aqueous medium were able to display excited states under light irradiation in the real systems (i.e., photosensitized dendrimer nanoaggregates) studied here. 

## 4. Conclusions 

The photocatalytic activity of G4-F nanoaggregates toward light-driven water photoreduction was achieved. The effect of the irradiation time on G4-F nanoaggregate systems was comprehensively characterized in order to gain a better understanding of the stability and photocatalytic activity over time, as well as the light-harvesting ability of these systems. Notably, the adopted arrangement of dendrimer nanoaggregates provided an adequate self-assembled scaffold for supporting fluorescein entities, thus contributing to the eventual stabilization of the formation of the excited state and facilitating the charge transfer processes. Interestingly, this study also demonstrated the effective performance of tandem G4-F nanoaggregates/Pt-ascorbate nanoparticles in the process of water photoreduction. Therefore, we believe that these studies can be considered a promising approach for the technological development (e.g., reduction or discoloration of dye wastewaters) of photoactive nanoaggregate materials for the visible-light-driven production of H_2_. Theoretical analyses helped to explain the results obtained by the different experimental techniques used in this study. It seems that the probability of the formation of excited electronic states of fluorescein entities induced by light is crucial for the photoproperties of the nanoaggregates. Finally, it should be noted that more experiments are in progress, in order to adequately complement the obtained preliminary results for this proof of concept.

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
