# Peer review of "Exploring the Effect of the Irradiation Time on Photosensitized Dendrimer-Based Nanoaggregates for Potential Applications in Light-Driven Water Photoreduction"

_nanomaterials, 2019, doi:10.3390/nano9091316_

Round 1
Reviewer 1 Report
In this contribution the authors present a dendrimer-based nanoaggregate for water photoreduction. Fluorescein units are covalently bound to a PAMAM dendrimer and use as photosensitizer.
First of all, the rational behind the choice of the system fluorescein + L-ascorbate + Pt-ascorbate nanoparticles + methyl viologen is not clearly explained, in particular the choice of the fluorescein as photosensitizer.
The authors provide a characterization of the aggregates formed by this modified PAMAM and the evolution of their properties under irradiation. The author observed drastic changes in the fluorescein spectra (UV-vis, FITR) after 12 h of irradiation, can they provide information about the stability of the aggregates without irradiation?
Although the author characterized the G4-F NG, no characterization of the nano-aggregates in presence of the metallic NP is provide. This characterization before and after irradiation is crucial, especially as the author observed H2 production in presence of G4-F NG and nothing with free fluorescein.
Besides, the study of the emission properties of the fluorescein (and eventual variations) is fundamental to go further in the description of this system.
From an aesthetic point of view, figures 5, 6 and 7 are too big and can be improved. Labels are not always visible (fig. 1, 3, 5, 6).
Author Response
Comments Academic Editor:
COMMENTS from Academic Editor :
"I think that authors should better investigate on the origin of H2 because I
am not sure that it derived only from water reduction. In my opinion H2 could
derive also by the degradation of PAMAM-fluorescein nanoaggregates."
Dear Editor, thank you for your invaluable contribution. Attending your concern, previously, we carried out diverse experiments to prove the stability and possible photoactivity of dendrimer aggregates at different times of irradiation and under the same experimental conditions of those used in water, using methanol and ethanol as solvents. Importantly, a significant generation of in situ hydrogen was not detected in either methanol or ethanol. Additionally, UV-visible spectra of our system did not show notable changes for the same periods of time in both solvents. While it is true that we cannot preliminary rule out that, the decomposition of nanoaggregates contributes to the generation of hydrogen in an aqueous medium, it could be said that this occurs only in the latter medium.Additionally, some paragraphs in this paper needs to be modified to avoid too much repetition. Please revise this paper based on enclosed report.
Reviewer #1
In this contribution the authors present a dendrimer-based nanoaggregate for water photoreduction. Fluorescein units are covalently bound to a PAMAM dendrimer and use as photosensitizer.
Dear reviewer, many thanks for your appreciable contributions and suggestions. We are very grateful for your comments. So considering your main concerns about this manuscript, the answers and/or comments are given below.
First of all, the rational behind the choice of the system fluorescein + L-ascorbate + Pt-ascorbate nanoparticles + methyl viologen is not clearly explained, in particular the choice of the fluorescein as photosensitizer.
You are right; recently, we have been exploring this research field (i.e., hydrogen evolution) and apparently, the conditions for the in situ generation of hydrogen from an aqueous medium can be generalized as follows:
A photosensitizer o photocatalyst, an electron donor, an electron relay and an active site for the reduction of water molecules (i.e., co-catalyst).
As photosensitizers can be found ruthenium, iridium complexes as classical systems for these purposes
As electron donor: EDTA, TEA, sodium citrate, etc,.
As electron relay are extensively used methyl or ethyl viologen
Finally, as co-catalyst Pt nanoparticles or Nickel complexes are commonly used.
We try to propose some variations for these systems, namely, the photosensitizer and the electron donor (in this case L-ascorbate).
The choice of fluorescein is mainly based on the absorption region of this molecule (visible range), the well-documented photoproperties of this molecule and, from the economic perspective, the low costs of it compared with other chromophores/fluorophores. Interestingly, the next work that we are conducting is the incorporation gold nanoparticles (specifically nanorods) in order to study the possible effect of improving the photoproperties of fluorescein in these experiments.
All the mentioned above is strongly supported by the extensive literature revision carried out by our group for different molecular systems.
The authors provide a characterization of the aggregates formed by this modified PAMAM and the evolution of their properties under irradiation. The author observed drastic changes in the fluorescein spectra (UV-vis, FITR) after 12 h of irradiation, can they provide information about the stability of the aggregates without irradiation?
You are right; these experiments were also previously conducted. From the information obtained from UV-visible spectra the dendrimer nanoaggregates are stable in the darkness and under normal light , at least if we consider the same time periods studied for the conducted experiments for water photoreduction.
Although the author characterized the G4-F NG, no characterization of the nano-aggregates in presence of the metallic NP is provide. This characterization before and after irradiation is crucial, especially as the author observed H2 production in presence of G4-F NG and nothing with free fluorescein.
You are right; TEM with Pt nanoparticles could be useful in this case. As you suggest the presence of Pt nanoparticles should condition the behavior of the aggregates. In this first approach, we think the characterization of dendrimer nanoaggregates without Pt nanoparticles, L-ascorbate, methyl viologen, etc would be an adequate primary way to achieve a better understanding of the light response of this type of material. Note that your concerns were addressed in previous experiments; however, we could not obtain satisfactory images for these samples (an example of micrograph is shown below). The rational design of further experiments are being carried out in order to provide a viable alternative to assess the effect of the other molecules in the surrounded environment of nanoaggregates. These studies are conducted to get information on the effect of dendrimer generation, wavelength contributions, fluorescence phenomena, cycles of irradiation, incorporation of metal nanoparticles and a deeper theoretical analysis, among others. It is expected that these obtained results can be published and help complement the results of this preliminary work.
Besides, the study of the emission properties of the fluorescein (and eventual variations) is fundamental to go further in the description of this system.
You are right; these experiments are being carried out to complement the results that are presented in this manuscript. As you can guess, the experiments as well as the analysis of them require much more time than it is available to give you a satisfactory response. As mentioned above, the fluorescence studies will be addressed in a further work.
From an aesthetic point of view, figures 5, 6 and 7 are too big and can be improved. Labels are not always visible (fig. 1, 3, 5, 6
You are right; your suggestions are very valid and have been considered for the new version.

Reviewer 2 Report
I suggest this paper to be published in the present form.
Author Response
Dear reviewer, many thanks for your very appreciable comments.
Round 2
Reviewer 1 Report
I thank the authors for their answers and explanations and I think a corrected/modified version of their manuscript is suitable for publication.
Author Response
Dear Reviewer, thank you for all your suggestions. According to your concerns, some sentences were added in the Conclusions section (marked in yellow)